# Application of *Bacillus coagulans* in Animal Husbandry and Its Underlying Mechanisms

**DOI:** 10.3390/ani10030454

**Published:** 2020-03-09

**Authors:** Yuanhao Zhou, Zihan Zeng, Yibin Xu, Jiafu Ying, Baikui Wang, Muhammed Majeed, Shaheen Majeed, Anurag Pande, Weifen Li

**Affiliations:** 1Key Laboratory of Molecular Animal Nutrition of the Ministry of Education, Institute of Feed Science, College of Animal Sciences, Zhejiang University, Hangzhou 310058, China; zyh17767072477@163.com (Y.Z.); 21817016@zju.edu.cn (Z.Z.); 21817002@zju.edu.cn (Y.X.); 21817401@zju.edu.cn (J.Y.); wangbaikui@zju.edu.cn (B.W.); 2Sami Labs Limited, Bangalore, Karnataka 560058, India; mail1@samilabs.com; 3Sabinsa Corporation, East Windsor, NJ 08520, USA; shaheen@sabinsa.com (S.M.); anurag@sabinsa.com (A.P.); 4Sabinsa Corporation, Payson, UT 84651, USA

**Keywords:** *Bacillus coagulans*, probiotics, nutrient metabolism, immunomodulation, antioxidant

## Abstract

**Simple Summary:**

Probiotics, a kind of feed additive, are widely used in animal husbandry and the effects are quite positive. Many strains of *Bacillus* spp. are currently used as probiotic dietary supplements in animal feeds. *Bacillus coagulans*, a probiotic with good application prospects, piqued our strong interest. In this review, information on *Bacillus coagulans* in scientific research and production practices is summarized.

**Abstract:**

In recent decades, probiotics have attracted widespread attention and their application in healthcare and animal husbandry has been promising. Among many probiotics, *Bacillus coagulans* (*B. coagulans*) has become a key player in the field of probiotics in recent years. It has been demonstrated to be involved in regulating the balance of the intestinal microbiota, promoting metabolism and utilization of nutrients, improving immunity, and more importantly, it also has good industrial properties such as high temperature resistance, acid resistance, bile resistance, and the like. This review highlights the effects of *B. coagulans* in animal husbandry and its underlying mechanisms.

## 1. Introduction

Probiotics are defined as “live microorganisms that, when administered in adequate amounts, confer a health benefit on the host” [1]. Furthermore, microorganisms that can be classified as probiotics should meet the following requirements: (1) to survive in the gastrointestinal environment, that is, to be acid and bile resistant; (2) to adhere to intestinal epithelial cells; (3) to grow rapidly, colonize the intestinal tract, to persist there and then leave the hosts; (4) to stabilize the intestinal microbiota; (5) to have no signs of pathogenicity; and (6) to maintain vitality both in food and pharmaceutical processes [2,3]. Probiotics have attracted more and more interest from animal nutritionists and livestock producers owing to their benefits to the gastrointestinal tract [4], and their role as one of the best alternatives to antibiotics in animals including poultry and fish [5]. Probiotics possess the function of promoting growth [6] and can be used as growth enhancers [7]. In addition, numerous studies have shown that probiotics have antimicrobial activity [8,9,10], which allows their consideration as alternatives to antibiotics. Probiotics could prevent or inhibit the proliferation of pathogens, suppress the production of virulence factors and perform beneficial functions as biofilms [11].

There are several types of probiotics used in the food sector, medical care and animal husbandry, such as *Lactobacillus*, *Bifidobacterium*, *Bacillus subtilis*, and others. Among them, *Lactobacillus* and *Bifidobacterium* are the most commonly used probiotic preparations [12]. However, the majority of these strains are not resistant to extreme temperatures, as well as stomach acid, digestive enzymes and bile salts [13]. Moreover, it is necessary to characterize other types of probiotics because of the technological difficulties with the delivery mechanism with the use of *Lactobacillus* and *Bifidobacterium*. Due to the non-sporulating characteristic of these genera, freeze drying is required, which, in turn, reduces the shelf life of powdered probiotic preparations and adversely affects their viability [14].

It is worth pointing out that many strains of *Bacillus* spp. are currently used as probiotic dietary supplements [15]. One of the reasons is that spore-forming probiotics are highly resistant to extreme environmental conditions [16]. *Bacillus* spp. have a higher acid tolerance and better stability in the process of heat treatment and low temperature storage compared to other probiotics [17]. *Bacillus coagulans* (*B. coagulans*) is a Gram-positive, facultative anaerobic, nonpathogenic, spore-forming and lactic acid-producing bacterium [18]. It is known as ‘the king of probiotics’ due to its high stability in the gastrointestinal tract, non-toxic effects, as well as high but not yet fully understood pharmacological activity [19]. *B. coagulans* is resistant to temperature; the perfect growth temperature for *B. coagulans* is 35 to 50 °C and the optimum growth pH is 5.5 to 6.5 [18,20]. Furthermore, spores of *B. coagulans* possess strong resistance, revival and stability, which can be activated in the acidic environment of the stomach and begin to germinate and proliferate in the intestinal tract [21]. Spores can accommodate to the intestinal low oxygen environment and reach the gastrointestinal tract smoothly so as to play the role of lactic acid bacteria (LAB) in the intestinal tract [21]. It has been found that the survival of the spores of *B. coagulans* during simulated digestion is 92% [22]. Once germinated, *B. coagulans* could produce a bacteriocin called coagulin, which has activity against a broad spectrum of enteric microbes [23], which is one of the mechanisms for *B. coagulans* to exert antibacterial activity. The factors previously listed confer *B. coagulans* many advantages over other LAB strains. It is worth pointing out that *B. coagulans* lacks the ability to adhere to the intestinal epithelium unless long-term administration is maintained. *B. coagulans* will be completely eliminated in four to five days [24]. Owing to this characteristic, *B. coagulans* may need long-term administration in order to play the role of a probiotic. For example, it has been shown that, *B. coagulans* spores, when administered at a level of 10^9^ spores/day for 30 days, could affect the intestinal microbiota of rats [25].

The discovery of *B. coagulans* dates back to 1915, when it was first found in curdled canned evaporated milk. It was described by the Iowa Agricultural Experiment Station [26], and isolated for the first time in 1932 [27]. At the beginning of the discovery, *B. coagulans* was known as *Lactobacillus sporogenes* and considered as a promising probiotic candidate, possessing the common characteristics of both *Bacillaceae* and *Lactobacillaceae* [28]. Nowadays, there are several strains of *B. coagulans* that have been used in animal feeds and research, such as *B. coagulans* ATCC 7050 [21] and commercial strains like *B. coagulans* MTCC5856 [29]. The application of *B. coagulans* in commercial feed and pet supplements has been supported by the stability in processed food [30] as well as phenotypic and genetic consistency for this strain [29]. Moreover, the safe application of *B. coagulans* has been supported by a toxicological safety assessment by Endres and colleagues [31]. They proved that *B. coagulans* is safe after conducting a one-year chronic oral toxicity study combined with a one-generation reproduction study [32].

## 2. The Application of *B. coagulans* in Animal Husbandry

Many studies have shown that the application of *B. coagulans* obtained good results in animal husbandry. For example, *B. coagulans* could decrease the diarrhea rate and improve the growth performance of piglets [33]. In addition, *B. coagulans* has been used widely in poultry production. It is well worth mentioning that *B. coagulans* has a growth-promoting effect on broiler chickens [34], possibly via improving the balance of intestinal microbiota to improve the feed conversion ratio [21]. Furthermore, *B. coagulans* not only displays a growth promoting effect in broilers, but also increases the activities of protease and amylase [35]. We know that protease and amylase play a crucial role in the fermentation of relative nutrients [35]. As for local chickens, such as the Guangxi Yellow chicken, the supplementation of *B. coagulans* could improve their growth performance and showed positive effects on meat quality [36]. Moreover, probiotics are also widely used in aquaculture. For example, *B. coagulans* could significantly improve the final weight, daily weight gain and relative weight gain of the shrimp [37]. In addition, a diet supplemented with *B. coagulans* had similar effects on a grass carp [38].

## 3. *B. coagulans* Promotes Nutrient Metabolism

One of the most beneficial effects of probiotics is that they could produce a large number of active enzymes in the process of metabolism and promote the absorption of nutrients to improve the feeding conversion efficiency. Secondly, probiotics could promote the synthesis and metabolism of proteins, vitamins and short chain fatty acids [39,40,41]. These metabolites exert many positive effects on energy metabolism.

*B. coagulans* could help digest carbohydrates and proteins once they are activated and germinated [42]. *B. coagulans* could improve growth performance and increase feed digestibility via producing amino acids and vitamins and secreting a-amylase, xylanase, protease and lipidase [34]. Furthermore, consumption of *B. coagulans* creates a healthy and efficient intestinal environment [43]. For example, when birds were fed a diet with *B. coagulans*, even though they were infected by *Salmonella enteritidis* later, their average body weight gain and the feed conversion ratio during the initial stage could still be improved [6]. Such a beneficial effect is probably owed to the promoted secretion of endogenous enzymes which improves the nutrient digestibility [35]. Moreover, *B. coagulans* also produces some exogenous enzymes and unknown growth-promoting factors which could increase the intestinal peristalsis and the feed digestibility [44,45]. *B. coagulans* produces α-galactosidase which strongly resists hydrolysis by protease [46]. α-galactosidase is an enzyme which catalyzes the removal of terminal nonreducing galactose residues from different substrates [46]. However, it is worth identifying which factor plays an important role in this nutrient metabolism process. In addition, it has been found that *B. coagulans* could decrease inflammation, resulting in the development of the absorptive area of the villi and the enhancement of nutrient absorption [47].

Probiotics offer numerous health benefits, including digestive health. Improving digestive health is associated with the more efficient absorption of crucial nutrients from the diet [48]. *B. coagulans* could help liberate digestion products, which are shorter peptides and more free amino acids than those produced by the control group, an in vitro model lacking probiotics [49]. In addition, *B. coagulans* also improves protein absorption and maximizes health benefits [48]. *B. coagulans* could promote nutrient metabolism via increasing the expression levels of genes related to nutrient absorption and transportation [33]. Prebiotics are defined as “non-absorbable food components that beneficially stimulate one or more of the gut-beneficial microbe groups and thus have a positive effect on human health” [50]. Probiotics have a cooperative phenomenon with prebiotics. *B. coagulans* could ferment with galactomannan, a kind of prebiotic, and produce short chain fatty acids, lactic acid and other compounds to play an important role in modulating intestinal microbiota [51]. In addition, *B. coagulans* could increase the population of beneficial bacteria to increase the benefits of prebiotics [52].

## 4. *B. coagulans* Regulates Immune Function

The gastrointestinal system is closely connected with the immune system [53]. The immune system identifies whether substances are harmless or potentially deleterious at the interface between the lamina propria and the enteric cavity [54]. Oral probiotics interact with gastrointestinal (GI) mucosa and gut associated lymphoid tissue (GALT) where more than 70% of immune cells are located [55]. Probiotics possess the power to influence the innate and adaptive immune system by interacting with several types of immunological cells such as B cells, T cells, regulatory T cells, dendritic cells, natural killer cells, monocytes and macrophages along the mucosa [56,57]. There are several types of interactions between intestinal microorganisms and the host cells and tissues, including via bacterial cell wall components and secreted metabolites [58].

*B. coagulans* could modulate an immune system, stimulate the rapid recovery of normal intestinal microbiota and possess antiviral activity [59]. *B. coagulans* could affect the intestinal microbiota through transient proliferation in intestines [25]. It could modify the ecology of gastrointestinal microorganisms by supplementing the quantity of the desirable obligate microorganisms and antagonizing pathogenic microorganisms in pigs [1]. In the matter of reaction with the immune system, the immunomodulatory function of *B. coagulans* is closely interrelated with cytokines. It has been found that when human blood cells are exposed to viruses, such as adenovirus and influenza A, *B. coagulans* treatment could significantly increase the production of cytokines, such as interleukins(IL-6, IL-8), tumor necrosis factor(TNF-α), and interferon(IFN-γ) [47]. *B. coagulans* exhibits an immunomodulatory effect by reducing IL-8 and increasing IL-10 secretion and it could also modulate the immune system to resist the inflammation caused by lipopolysaccharide [22]. As for the application in animal husbandry, the continuous addition of *B. coagulans* helps to reduce the host immune inflammatory response induced by *Salmonella enteritidis* [6]. These results indicate that *B. coagulans* is crucial for inducing both inflammatory mediators and anti-inflammatory cytokines, which could help the immune system to protect the host against infections and minimize inflammation-related tissue injury.

As for primary mechanisms of immunomodulation, there are two potential factors in this process. On the one hand, the most direct interaction between microorganisms and the host’s cells are the components of outer layers of bacterial cell wall and the receptors on immune cells [58]. The receptors could interact with lipopolysaccharide, flagellin, lipoteichoic acid and lipoproteins, which are present on the exterior surface of bacteria [54]. In addition, toll-like receptors (TLR) play an important role against pathogens via activating the transcription factor NF-κB signaling pathway and inducing inflammatory cytokines, such as TNF-α, IL-6, IL-1β, IL-8, and IL-12, IFN and co-stimulatory molecules [60]. Gram-positive bacteria, such as *B. coagulans*, could combine with the TLR-2 receptor via the teichoic acid and lipoteichoic acid [61], and the chemical composition of the bacterial cell wall plays an important role in regulating immune function [58]. It is worth mentioning that the teichoic acid of *B. coagulans*, which is a glycerophosphate polymer, substituted with glucose and galactose but lacking amino acid substituents, contains a higher lipid content than the majority of Gram-positive bacteria [62]. In this way, *B. coagulans* possesses the ability to trigger complex beneficial immune reactions [63]. In a study by Jensen et al. [58], they tried to figure out whether the inactivated *B. coagulans* possesses immune activating and anti-inflammatory functions or not, and they finally demonstrated that it could activate immune cells and alter the production of chemokines and cytokines. Through this research, we conclude that the components of bacterial cell walls are crucial for immunomodulation. On the other hand, epithelial cells play a crucial role in assimilating nutrients and forming a mucosal barrier which could protect tissue from damage [64]. Probiotics could stimulate the mucosal defense to produce antimicrobial peptides [65]. In addition, commensal bacteria could modulate the immune system via secreting certain bioactive compounds, which possess unique health benefits for the host. Benson and colleagues [66] found that the maturation of antigen-presenting cells could be supported by metabolites of *B. coagulans* in vitro. In addition, the metabolites of *B. coagulans* possess strong immune modulating properties and anti-inflammatory effects in vitro [54]. These research studies suggest that *B. coagulans* may exert its effects by secreting certain bioactive compounds to modulate the immune system.

## 5. *B. coagulans* Possesses Antioxidant Ability and Alleviates the Toxicity of Heavy Metals

The intestinal microbiota plays a crucial role in protecting its host from pathogenic microbes via occupying adhesion sites, producing antibacterial substances and consuming nutrients [67]. When the intestinal microbiota becomes abnormal, harmful microorganisms proliferate excessively and eventually cause significant oxidative stress [68]. Oxidative stress means that the intracellular levels of oxygen radicals increase, resulting in damage to lipids, proteins, and DNA [69], and it is related to many pathological conditions, as well as the increased levels of reactive oxygen species (ROS) and lipid peroxidation [70,71]. The production of high-level free radicals in the intestine will exert cytotoxic effects on the membrane phospholipids of the intestinal epithelial cells [72].

Probiotics could produce different kinds of antioxidative metabolites, such as glutathione (GSH), butyrate, and folate [73]. In addition, probiotics are important factors which affect the oxidative status of the intestine via exhibiting direct antioxidant properties and inducing the intrinsic human signaling antioxidant defense [72,74]. *B. coagulans* could alleviate the oxidative stress via increasing the activities of myeloperoxidase (MPO) and anti-superoxide anion free radical (AFASER), decreasing the content of malondialdehyde (MDA), regulating the transcriptional regulation levels of antioxidant enzymes and Nrf2-Keap1 signaling molecules [75].

Numerous studies pointed out that several types of heavy metals, such as cadmium and mercury, caused oxidative stress via inducing the production of ROS [76,77]. One of the most important reasons is that heavy metals influence some key enzymes of the antioxidant system, like superoxide dismutase (SOD) [78]. Several probiotics which belong to Gram-positive bacteria could release this oxidative stress by binding heavy metals. *Bacillus* spp. possess a high absorptive capacity owing to the high content of peptidoglycan and teichoic acid of their bacterial cell walls [79]. *B. coagulans* could protect the host from the oxidative stress caused by mercury [76]. Research into acute cadmium toxicity treated with synbiotics (consisting of *B. coagulans*, *Lactobacillus plantarum* and inulin) resulted in increased activity of antioxidant enzymes and a reduced level of cadmium in the tissues [77]. The antioxidant property of *B. coagulans* could be used in animal husbandry to prevent heavy metal poisoning.

## 6. Conclusions

*B. coagulans* is a spore-forming and lactic acid-producing bacterium which possesses the capacity to improve the feed conversion ratio and the balance of intestinal microbiota, ultimately improving the growth performance of animals. It may also regulate immune function, help the immune system to protect against infections and minimize inflammation-related tissue injury. In addition, *B. coagulans* possesses antioxidant ability and alleviates the toxicity of heavy metals. Thus, *B. coagulans* plays a crucial role as a microbial agent and has a beneficial role as a feed additive. The mechanism of the effects of *B. coagulans* in animal husbandry warrants further investigation.

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
