# Peer review of "Application of Bacillus coagulans in Animal Husbandry and Its Underlying Mechanisms"

_animals, 2020, doi:10.3390/ani10030454_

Round 1

Reviewer 1 Report

Authors revised the manuscript according to the instructions.

Author Response

Thank you very much for your comments on my manuscript. Your suggestions and ideas will be of great help to our later writing!I wish you every success in your research!

Reviewer 2 Report

Overall, the authors have answered the questions I have included adequately but have not taken steps to improve so much the quality of English in the manuscript, which could greatly improve the manuscript. The only points revised were the ones I suggested for correction, but unfortunately I am not reviewing for the sake of correcting the grammar, but rather to improve content and correctness as a review paper. It would be suggested to have the manuscript to be checked by an English speaker.

A few points are the following:

  • Simple Summary:
    • Line 19: “a kind of feed additive”
    • Line 16: Remove “application”
    • Line 17: “As a probiotic”
    • Line 18: “some research” -> “information”
  • Abstract:
    • Line 21: “has brought great benefit to us” -> “has been promising”
    • Line 22: “hot spot” -> key player
    • Line 23-25: It has been demonstrated to be involved in regulating the balance of the intestinal microbiota, promoting metabolism and utilization of nutrients, improving immunity, and more importantly, it also has good industrial properties such as high temperature resistance, acid resistance, bile resistance, and the like.
  • Introduction
    • Line 42: "Which allow their consideration as alternatives to antibiotics"
    • Line 45: “Food sector”
    • Line 51: remove “technology”
    • Line 57: “…compared to other probiotics”
    • Line 63: “resurrection” -> "revival"
    • Line 81: “research”
    • Line 86: “For a one-year”
    • Line 91: “of piglets”
    • Line 93: “to improve”
    • “We need to know that protease and amylase played a crucial role in the fermentation of relative nutrients[36]” -> does not contribute to the section. Give more details and (e.g. relative nutrients such as?)
    • Line 97: “supplementation”: “…improved their growth… “
    • Line 100: “Besides” -> “In addition,”

I will not continue to detail line-by-line corrections for English usage, but I do request the authors for English usage correction for it to be fit for publication. There are a few more that require correction in the manuscript and once they are improved, I believe this can be accepted for publication in Animals.

Author Response

Thank you very much for your comments. Your comments are of great help to our manuscript. We have revised it according to your comments and asked my English professional friend to help with the revision. The point to explain is, “We need to know that protease and amylase played a crucial role in the fermentation of relative nutrients[36]”, this sentence is another reviewer suggested to add to explain the role of the enzymes mentioned earlier. Thank you again for your constructive comments on our manuscript.

This manuscript is a resubmission of an earlier submission. The following is a list of the peer review reports and author responses from that submission.

Round 1

Reviewer 1 Report

Authors performed corrections of the manuscript according to my previous reviosion points. Please concern citing new review articles. Native speaker corrections od English recommended (is/are, etc).

Reviewer 2 Report

Dear Authors,

your text is interesting. However, in my opinion, despite a lot of information, this is not a review article that meets the guidelines of Guidelines for writing a Review Article, http://www.plantscience.ethz.ch/education/Masters/courses/Scientific_Writing. Content is popular, citations do not make an article a review article. I suggest to completely rewrite this article, which will be beneficial for the reader and for the authors of the article.

I suggest that authors could read review articles like: Ashmann T-L. & C. J. Majetic (2006). Genetic constraints on floral evolution: a review and evaluation of patterns. Heredity 96: 343 – 352 lub Kessler A. & I. T. Baldwin (2002). Plant responses to insect herbivory: The emerging molecular analysis. Annual Review of Plant Biology 53: 299 – 328
A.M. Lindig, P.D. McGreevy, A.J. Crean (2020). Musical Dogs: A Review of the Influence of Auditory Enrichment on Canine Health and Behavior. Animals, 10(1), 127;doi:10.3390/ani10010127
I. Rychlik (2020). Composition and Function of Chicken Gut Microbiota. Animals 2020, 10, 103; doi:10.3390/ani10010103.

Reviewer 3 Report

Zhou et al. presents a summary of the available scientific evidence for use of Bacillus coagulans in animal husbandry while providing mechanistic insights into these probiotic characteristics. Overall, the manuscript is concise and organized. I believe that the aim has been reached but that the paper requires some minor edits as to the grammar and usage, and suggestions to make it better as a review paper as follows:

TITLE: Suggestion: “Application of Bacillus coagulans in animal husbandry and its underlying mechanisms”. The advances in research seems quite redundant for the title. Simple Summary: Line 19: “caused” -> “piqued”, “some researches” -> “some research” Abstract: Line 21: “…widespread attention and their application in health care…” Line 26: “…temperature resistance, acid resistance, bile resistance, and the like.” How about for transient colonization? One of the most important aspects of probiotic administration is regarded for this ability to be able to actually produce its beneficial effects to the host. As it was determined that it requires long-term administration and is completely eliminated in four to five days, it may be helpful for the authors to cite evidence or propose suggestions for further research regarding the appropriate time for administration in animal husbandry to be able to acquire the desired effects. This should then be added to Part 2, “The application of coagulans in animal husbandry” and connected to their ability to regulate immune function “B. coagulans regulates immune function”. Both are related in a way that if there isn’t a intimate contact establish between B. coagulans and the host cells in animal trials, it seems highly unlikely that it directly produces the immune function. Introduction: Line 37: “…of pathogenicity; and, (6) to maintain…” Line 39: “producers owing to their benefits to the gastrointestinal tract” Line 40: ”..and their role as one of the best alternatives…” Line 42: “…there were numerous studies that have shown…” Line 47: “et al.” -> “and others” Line 47: “Bifidobacteria” to “Bifidobacterium” Line 48-49: “However, they couldn’t resistant to extreme temperatures, as well as stomach acidy, digestive enzymes and bile salts”: This line suggests that in general, the other Lactobacillus, Bifidobacterium, Bacillus subtilis are all unable to resist extreme environments which is untrue, not for all. Again, this is a matter of strain/species specific context and must be revised to reflect so. Line 51: “Bifidobacteriua” -> “Bifidobacterium” Line 70: “…many advantages over other LAB strains…” Line 71: “… to the intestinal epithelium unless long-term administration is maintained” Would this be economical then for animal husbandry in the long term? How long is long-term administration? How can this be established as in the case for animal husbandry? This would be good to be mentioned in this section. Part 2. The application of coagulans in animal husbandry Line 89: “There were many studies that have shown…” Line 94: “mcirobiota” -> “microbiota” Line 96: “increased the activities of protease and amylase” Would be good to briefly expound on the what the effects of increasing these enzymes are contributing to in the animal health Line 100: “…relative weight gain of treatments…” Part 3: Line 88: “ coagulans promotes nutrient metabolism” Line 115-116: “Other scholars suggested that…” (delete there are) Line 115 and 119: “α-galactosidase” Line 122: “…that coagulans could decrease inflammation, resulting to the development of the absorptive area of the villi and enhancement of nutrient absorption.” Part 4. coagulans regulates immune function Line 148-149: How does this occur when it does not adhere to the epithelium (to promote a close contact with the host cells) as mentioned in the introduction? It would be good to help the readers understand this mechanism. Line 161: “…there were studies suggesting that…” Line 180: “…finally demonstrated that it could activate immune cells…” Line 184: “…which could protect tissue from damage.” Line 186: “…to produce antimicrobial peptides.” Line 188 and 190: italicize “in vitro” Line 190: “These research studies suggest that…” Part 5. B. coagulans possesses antioxidant ability and alleviate the toxicity of heavy metals Line 193 and 195: “mcirobiota” -> “microbiota” Line 202: “…could produce different kinds of…”